miR-381-3p contribution in mouse spontaneous abortion via targeting VEGFA

Ding Chao 1 dingchao@njucm.edu.cn
Liu Fukang 1
Shi Huayue 1
Zuo Jing 1
Bi Lei 1
Shao Longgang 1
http://orcid.org/0009-0002-9462-8183 Pan Yanqiu 2 taoqipanpan@126.com
1 School of Traditional Chinese Medicine & School of Traditional Chinese, Nanjing University of Chinese Medicine , Nanjing, Jiangsu , China
2 Chinese Medicine Department, Nanjing Maternity and Child Health Care Hospital , Nanjing, Jiangsu , China
Uversky Vladimir
Electronic publication date: 2025 Jun 24
Publication date: 2025
Volume: 13
Electronic Location ID: e19568
Received 2024 Sep 20; Accepted 2025 May 14
Copyright: © 2025 Ding et al.
Copyright year: 2025
Copyright holder: Ding et al.
License: This is an open access article distributed under the terms of the Creative Commons Attribution License, which permits unrestricted use, distribution, reproduction and adaptation in any medium and for any purpose provided that it is properly attributed. For attribution, the original author(s), title, publication source (PeerJ) and either DOI or URL of the article must be cited.
License URL: https://creativecommons.org/licenses/by/4.0/

Keywords: Recurrent spontaneous abortion, miR-381-3p, VEGFA/NF-κB pathway, Migration, Angiogenesis

Funding: University Natural Science Research Project of Jiangsu Province 19KJB360012 Science and Technology Project of Jiangsu Chinese Medicine Bureau MS2022036 Natural Science Foundation project of Nanjing University of Chinese Medicine XZR2020052 This study was supported by the University Natural Science Research Project of Jiangsu Province (19KJB360012), the Science and Technology Project of Jiangsu Chinese Medicine Bureau (MS2022036), and Natural Science Foundation project of Nanjing University of Chinese Medicine (XZR2020052). The funders had no role in study design, data collection and analysis, decision to publish, or preparation of the manuscript.

==============================
Background

Recurrent spontaneous abortion (RSA) affects 1–5% of pregnant women; however, the mechanisms underlying this condition remain unknown. Dysangiogenesis in the placenta is an essential factor in the pathogenesis of RSA. Studies have verified that microRNAs (miRNAs) are vital for RSA; however, their mechanism of action in regulating angiogenesis remains unclear. Therefore, we explored the contribution of key miRNAs that regulate angiogenesis in RSA.

Methods

The abortion mouse model was constructed by intraperitoneal injection of beta2-Glycoprotein I (β2-GPI). The abnormal expression of miRNAs in the placenta of the abortion mice was screened using miRNA-seq. Based on miRNA databases, miR-381-3p, which is highly expressed in abortion mice, may bind to vascular endothelial growth factor A (VEGFA). Subsequently, we investigated the effects of the miR-381-3p/VEGFA axis on the angiogenesis of vascular endothelial cells using real-time quantitative polymerase chain reaction, Transwell, wound healing, tube formation, western blotting, and dual-luciferase reporter system. Furthermore, an in vivo experiment was used to confirm miR-381-3p knockdown contribution in the abortion mouse model.

Results

miR-381-3p overexpression inhibited the migration and angiogenesis of C166 cells (a mouse vascular endothelial cell line), whereas miR-381-3p knockdown had the opposite effect. The dual-luciferase reporter system revealed that miR-381-3p bound to the VEGFA 3′ UTR, and VEGFA knockdown counteracted the beneficial effect of the miR-381-3p inhibitor on angiogenesis. An in vivo study demonstrated that miR-381-3p knockdown may reduce inflammation and damage to the placenta and fetus during abortion by activating the VEGFA/nuclear factor kappa B (NF-κB) pathway.

Conclusion

miR-381-3p may cause insufficient placental blood flow by inhibiting the VEGFA pathway and can be used as a potential therapeutic target for RSA.

Introduction

Recurrent spontaneous abortion (RSA) is defined as two or more consecutive miscarriages before 20 weeks of pregnancy (de Assis, Giugni & Ros, 2024; Yao et al., 2024). RSA affects an estimated 1–5% of women of reproductive age (Mou et al., 2022; Sonehara et al., 2024). In approximately half of the cases, the reasons for miscarriage were unclear (de Assis, Giugni & Ros, 2024). However, there is no single comprehensive theory that explains the pathogenesis of RSA (de Assis, Giugni & Ros, 2024; Han et al., 2025). It is widely accepted that the pathogenesis of RSA is related to endocrine problems, genetic illnesses, autoimmune diseases, and anatomical anomalies (Davalieva, Kocarev & Plaseska-Karanfilska, 2025; Han et al., 2025). Recent studies have linked RSA pathogenesis to endothelial dysfunction and demonstrated that promoting angiogenesis at the maternal-fetal interface can rescue RSA (Abu-Ghazaleh et al., 2023; Wang et al., 2024; Zhang et al., 2023).

Vascular endothelial growth factor (VEGF) is a well-known angiogenic factor crucial for various physiological and pathological processes (Pérez-Gutiérrez & Ferrara, 2023; Vemuri et al., 2024). VEGF principally acts on blood vessel endothelial cells to maintain the integrity and permeability of the vascular wall and directly boost the growth of the cardiovascular system (Pérez-Gutiérrez & Ferrara, 2023; Zeng & Fu, 2025). VEGFA, a member of the VEGF family, is extensively expressed in the placenta during pregnancy and is considered a critical factor in the growth and development of the fetus (Krog et al., 2024; Mrozikiewicz et al., 2024). It has been reported that VEGFA absence is a significant contributing factor to RSA pathogenesis (Krog et al., 2024; Liao et al., 2024). Consequently, maintaining the VEGFA level is beneficial to improving RSA (Liao et al., 2024; Sajjadi et al., 2020).

MicroRNAs (miRNAs), a class of small non-coding RNAs, consist of 20–24 nucleotides (Diener, Keller & Meese, 2024; Seyhan, 2024). miRNAs are post-transcriptional regulators of gene expression that work by either inducing mRNA degradation or blocking the translation of their target genes (Diener, Keller & Meese, 2024; Saliminejad et al., 2019). A growing body of research has established the significance of miRNAs in RSA pathogenesis (Patronia et al., 2024; Wang et al., 2024). For example, miR-184 is abnormally expressed in RSA and induces apoptosis of trophoblastic cells by targeting zinc finger matrin-type 3 (Zhang et al., 2019), and miR-200c can inhibit human extravillous trophoblast cell migration, invasiveness, and proliferation by regulating PI3K/AKT signaling in unexplained recurrent spontaneous abortion (Yue & Xu, 2024). Moreover, exosomal miR-146a-5p and miR-146b-5p generated from M1 macrophages inhibit trophoblast cell migration and invasion by targeting TNF receptor-associated factor 6 (Ding et al., 2021). These studies suggest that miRNAs are involved in RSA. Currently, most miRNA activity in RSA is limited to controlling trophoblast cells; however, miRNA-mediated endothelial cell functions involved in RSA have not been fully appreciated (Wang et al., 2024). Endothelial cells are indispensable for remodeling placental blood flow (Tan et al., 2024).

In this study, differentially expressed miRNAs were screened in the placentas of the control and the abortion mouse model using miRNA-seq. As previously reported (Jiang et al., 2024; Zhao et al., 2025), miRNA databases (StarBase, miRwalk, TargetScan, and miRTarBase) were used to predict the miRNAs bound to VEGFA. Based on Venn diagram analysis, the predicted miRNAs and upregulated miRNAs in the placenta of the abortion mice overlapped with miR-381-3p. Consequently, we investigated miR-381-3p contributions in endothelial cells’ angiogenesis and abortion mouse model, providing a potential target for RSA therapy.

Materials and Methods

Animal model and miRNA-seq

Eight-week-old female and male C57BL/6 mice were obtained from Shanghai Lingchang Biotechnology Co., Ltd. (Shanghai, China). The mice were kept in the Laboratory Animal Center of Nanjing Ramda Pharmaceutical Co., Ltd. (Nanjing, China) and lived in CP-4 cages (L × W × H: 290 × 178 × 160 mm). The mice density of each cage did not exceed six. The mice were allowed to eat and drink freely at a temperature of 24 ± 2 °C, relative humidity of 40–60%, and a light-dark cycle for 12 h. After acclimatization for 1 week, ten female mice in proestrus were randomly divided into two groups: Control and abortion. As previously reported (Ding & Lu, 2014), after carbon dioxide (CO2) anesthesia, each animal in the abortion groups was intraperitoneally injected with 10 µg beta2-glycoprotein I (β2-GPI) (Abcam) on the 1st day and immunized once on the 8th day. The female mice in the control group were intraperitoneally injected with normal saline. On the 18th day, the female mice were mated with male mice at a ratio of 1:2. The presence of pessaries was considered to be on the 0.5th day of pregnancy. On the 15th day of pregnancy, the animals were euthanized using CO2, and the placenta tissues of the two groups were frozen in liquid nitrogen. As previously described (Li et al., 2024), miRNA-seq was used to detect miRNA expression profiles in the placentas of the two groups (n = 3). miRNA-seq and analysis of differentially expressed miRNAs were conducted by Beijing Boao Crystal Code Biotechnology Co., Ltd. (Beijing, China).

The animal experiment was approved by the Institutional Animal Care and Use Committee of Nanjing Ramda Pharmaceutical Co., Ltd. (No. IACUC-20230208) and performed following the Animal Care Committee.

Cell culture

C166 cells (a mouse vascular endothelial cell line) were acquired from Beijing Baiou Bowei Biotechnology Co., Ltd. (Beijing, China). The cells were cultured in Dulbecco’s Modified Eagle Medium with 10% fetal bovine serum (Life Technologies, Carlsbad, CA, USA) and 1% penicillin/streptomycin in an atmosphere of 5% CO2 and 95% air at 37 °C.

Cell transfection

miR-381-3p mimics, miR-381-3p inhibitor, VEGFA siRNA, and negative control (NC) were synthesized by Jiangsu KeyGEN Biotechnology Co., Ltd. (KeyGEN BioTECH, Nanjing, China). They were respectively transfected into C166 cells for 24 h using Lipofectamine™ 3000 (Invitrogen, Waltham, MA, US) following the manufacturer’s instructions. All experiments were repeated thrice. The sequences used are listed in Table 1.

Table 1 The sequences of miR-381-3p mimics, inhibitor, and VEGFA siRNA.

Name	Sequence (5′→3′)	
miR-381-3p mimics	F: UAUACAAGGGCAAGCUCUCUGU	
R: ACAGAGAGCUUGCCCUUGUAUA	
mimics NC	F: UCACAACCUCCUAGAAAGAGUAGA	
R: UCUACUCUUUCUAGGAGGUUGUGA	
miR-381-3p-inhibitor	ACAGAGAGCUUGCCCUUGUAUA	
inhibitor NC	UCUACUCUUUCUAGGAGGUUGUGA	
VEGFA siRNA#1	F: CCAAAGAAAGACAGAACAATT	
R: UUGUUCUGUCUUUCUUUGGTT	
VEGFA siRNA#2	F: ACAUAGGAGAGAUGAGCUUTT	
R: AAGCUCAUCUCUCCUAUGUTT	
VEGFA siRNA#3	F: CGGAUCAAACCUCACCAAATT	
R: UUUGGUGAGGUUUGAUCCGTT	
siRNA NC	F: UUCUCCGAACGUGUCACGUTT	
R: ACGUGACACGUUCGGAGAATT	

Wound healing assay

A straight line was drawn in the 6-well plate using a 200 μL yellow sterile pipette tip when the cell fusion degree had reached 80–90%. The cell migration was photographed through an inverted microscope (Olympus, Shinjuku City, Tokyo, Japan) at 0 and 24 h following transfection.

Transwell assay

Transwell assay was used to measure C166 cell migration. The transfected cells were added to the upper chamber of the Transwell, and a complete medium was added to the 24-well plate. After 24 h of culture, the medium in each well was removed. The cells in the Transwell chambers were fixed with methanol for 30 min. The upper cells of the microporous membrane in the Transwell chamber were removed using a cotton swab, and the lower cells were stained with 0.1% crystal violet (Sigma, Kanagawa, Japan) for 20 min. The migratory cells were photographed and counted using an inverted microscope (Olympus, Shinjuku City, Tokyo, Japan).

Tube formation assay

The angiopoiesis of C166 cells was examined using a tube formation assay. The cells were digested with 0.25% trypsin-EDTA (KeyGEN BioTECH, San Francisco, California, US) 24 h after transfection. Briefly, 200 µL of cell suspension (4 × 105 cells/mL) was added to per well in a 96-well plate coated with Matrigel (BD Biosciences, San Jose, CA, USA). After incubation at 37 °C for 6 h, angiogenesis was observed using an IX51 microscope (Olympus, Shinjuku City, Tokyo, Japan). The total branch points and capillary length were counted in three random fields of view using Gel-Pro32 software.

Dual-luciferase reporter assay

The wild-type (WT) and mutant-type (MUT) VEGFA sequences were designed following the binding sites of miR-381-3p and VEGFA 3′ UTR predicted by the StarBase database (https://rnasysu.com/encori/). VEGFA WT and MUT sequences (the 200 bp upstream and downstream of chr17:46017343-46017348) were inserted into pmirGLO vectors to construct pmirGLO-VEGFA-WT and pmirGLO-VEGFA-MUT recombinant vectors. The Dual-Glo® Luciferase Assay System (Promega, Madison, WI, USA) was used to determine the reporter activities. In a 96-well black plate, pmirGLO-VEGFA-WT or pmirGLO-VEGFA-MUT was co-transfected with the miR-381-3p mimic or NC into 293T cells using Lipofectamine™ 3000 (Invitrogen, Waltham, MA, USA). After transfection for 48 h, 50 µL of lysed cells were added to each well. Afterward, the Dual-Glo® Luciferase Reagent and Dual-Glo® Stop & Glo® Reagent were added to each well in turn. The fluorescence densities were detected by a Tecan Spark microplate reader (Tecan, Männedorf, Switzerland). The luciferase activities were calculated using the ratio of firefly/renilla activities.

miR-381-3p knockdown in vivo

As described in the animal model construction above, 15 eight-week-old female and male C57BL/6 mice in proestrus were randomly divided into three groups: Control, abortion, and abortion+shmiR-381-3p. On the 9th day of β2-GPI treatment, each animal in the abortion and abortion+shmiR-381-3p groups was intraperitoneally injected with 2 × 1011 vg of adeno-associated virus (AAV)-shNC or AAV-shmiR-381-3p synthesized from Guangzhou PackGene Biotechnology Co., Ltd. The female mice in the control group were intraperitoneally injected with normal saline. On the 18th day, female mice were mated with male mice. On the 15th day of pregnancy, the mice were euthanized by CO2, and the weight of the fetus and placenta was counted and photographed. The placenta was carefully divided into two parts: One was fixed with 4% paraformaldehyde, and the other was frozen with liquid nitrogen and then stored at –80 °C.

Total RNA extraction and real-time quantitative polymerase chain reaction (RT-qPCR)

The total RNA of the cells and placentas was extracted with TRIzol Reagent (Invitrogen, Waltham, MA, USA). The purity of the total RNA was determined using an ultraviolet-visible spectrophotometer (UV-2450; SHIMADZU, Kyoto, Japan). PrimeScriptTM RT master mix (Takara, Shiga, Japan) was added for reverse transcription of total RNA. For miRNA analysis, the first strand of miRNA cDNA was obtained using the stem-loop method. SYBR Green qPCR Mix (Takara, Shiga, Japan) was used to examine the gene expression. The PCR reaction procedure, including holding stage (95 °C for 5 min), 40 cycles (95 °C for 20 s, 60 °C for 20 s, and 72 °C for 20 s), and melt curve stage (95 °C for 15 s, 60 °C for 1 min, and 95 °C for 15 s), was performed using the LightCycler480 II real-time fluorescence quantitative PCR instrument (Roche, Basel, Switzerland). GAPDH or U6 was used as an internal reference. The relative expression levels of miR-381-3p and Vegfa were calculated using the 2−ΔΔCt method. Three biologically independent assays were performed. The primer sequences of miR-381-3p, Vegfa, Gapdh, and U6 were synthesized by Shanghai GenePharma Co., Ltd. (Shanghai, China), as listed in Table 2.

Table 2 The primer sequences of the genes.

Name	Sequence (5′→3′)	
Vegfa	F: CCACGACAGAAGGAGAGCAGAA	
R: TCTCAATCGGACGGCAGTAGC	
miR-381-3p	F: CGCGTATACAAGGGCAAGCT	
R: AGTGCAGGGTCCGAGGTATT	
miR-381-3p	GTCGTATCCAGTGCAGGGTCCGAGGTATTCGCACTGG	
RT	ATACGACACAGAG	
Gapdh	F: AAGGTCGGTGTGAACGGATT	
R: TGAGTGGAGTCATACTGGAACAT	
U6	F: CTCGCTTCGGCAGCACA	
R: AACGCTTCACGAATTTGCGT	

Western blotting

The cell and placenta samples were treated with EBC buffer (Roche Diagnostics, Basel, Switzerland, USA) and 1 mM phosphatase inhibitor mixture II (Sigma-Aldrich, St. Louis, MO, USA). SDS-PAGE was performed by transferring proteins onto nitrocellulose membranes. Subsequently, 5% non-fat milk was used to block the membranes. The membranes were then incubated overnight at 4 °C with primary antibodies, including VEGFA (19003-1-AP; Proteintech, Rosemont, IL, USA, 1:500), p65 (ab16502; Abcam, Cambridge, USA, 1:1,000), p-p65 (ab76302; Abcam, Cambridge, USA, 1:1,000), and GAPDH (ab9485; Abcam, Cambridge, USA, 1:2,000). After rinsing with 1× TBST; the membranes were incubated with the respective secondary antibodies conjugated with horseradish peroxidase for 1 h at room temperature. The protein bands were visualized using Immobilon™ Western Chemiluminescent HRP Substrate (Cat. No.: WBKLS0500; Millipore Corporation, Burlington, MA, USA), and the images were captured on the visualization instrument Tanon-5200 (Tanon, Xinjiang, China).

Hematoxylin-eosin (H&E) staining and immunohistochemistry (IHC) assay

On the 15th day of pregnancy, the placenta samples were fixed in a 4% paraformaldehyde solution and embedded in paraffin. The paraffin-embedded placental tissues were sectioned to a thickness of 4 μm, deparaffinized with xylene, and rehydrated in a graded series of ethanol. H&E staining was performed, and antigen retrieval was conducted by microwaving in the citric acid buffer. The sections were incubated with antibodies against CD31 (ab182981; Abcam, Cambridge, USA, 1:500), rinsed, and incubated with a secondary antibody for 60 min at room temperature. Three randomly selected visual fields were analyzed using Image-Pro Plus software (version 6.0).

Enzyme-linked immunosorbent assay (ELISA)

The blood of each mouse was obtained from the eye socket and centrifuged at 3,000×g, 4 °C for 20 min. The concentrations of serum total proteins were detected by a BCA assay kit (KeyGEN BioTECH, Nanjing, China). The levels of interleukin (IL)-10, tumor necrosis factor α (TNF-α), homocysteine (HCY), annexin V (ANX-V), and methylene tetrahydrofolate reductase (MTHFR) were measured using an ELISA Kit (Shanghai Enzyme-linked Biotechnology Co., Ltd., Shanghai, China) under a Tecan Spark microplate reader (Tecan, Männedorf, Switzerland).

Statistical analyses

All data are presented as mean ± standard deviation (SD) of at least three biological replicates per experiment. The data were statistically analyzed using the Statistical Package for the Social Sciences (SPSS) software (version 22.0). A one-way analysis of variance was used to analyze differences among groups using the LSD method. A P < 0.05 indicated statistical significance.

Results

miR-381-3p inhibited the migration and angiogenesis in C166 cells

To explore potential miRNAs that regulate angiogenesis in the placental tissue of RSA, miRNA-seq was used to identify differentially expressed miRNA profiles in normal and abortion-pregnant mice. The results demonstrated that 133 miRNAs were upregulated, and 24 miRNAs were downregulated in the placental tissue of the abortion group than the normal group (Figs. 1A, 1B and Table S1). Among the upregulated miRNAs, miR-381-3p bound to VEGFA with a higher binding score (Fig. 1C), according to multiple miRNA databases (StarBase, miRwalk, TargetScan, and miRTarBase) (Jiang et al., 2024; Zhao et al., 2025). This suggests that miR-381-3p may affect placental angiogenesis by regulating VEGFA. The miRNA-seq raw data were uploaded to the Gene Expression Omnibus Datasets (Accession: GSE286463).

Figure 1 MiRNA expression profile in the placentas of the control and abortion mice and binding prediction of miRNAs and VEGFA.

(A) Volcano plot assay. (C) Heatmap assay. (B) miRNA databases predicted the miRNAs binding to the VEGFA 3′ UTR.

Subsequently, we examined whether miR-381-3p regulated angiogenesis in vitro. As presented in Fig. 2, miR-381-3p mimics significantly inhibited the migration of C166 cells (P < 0.001) (Figs. 2A–2C) and reduced branch points and capillary length during angiogenesis (P < 0.01) (Figs. 2A, 2D and 2E), compared with the NC group. However, miR-381-3p inhibitor revealed the opposite effect (Figs. 2C–2G). These findings demonstrated that miR-381-3p suppressed the migration and angiogenesis in C166 cells.

Figure 2 miR-381-3p inhibited migration and angiogenesis of C166 cells.

(A) Migration and angiogenesis of C166 cells were measured using Transwell (magnification: 20×), wound healing (magnification: 10×), and tube formation (magnification: 10×) assays. (B) Migration number of C166 cells. (C) Migration distance of C166 cells. (D) Branch points in angiogenesis of C166 cells. (E) Capillary length in angiogenesis of C166 cells. Data are presented as mean ± SD (n = 3). *P < 0.05, **P < 0.01, and ***P < 0.001.

miR-381-3p suppressed the activation of the VEGFA/NF- κB pathway

Activation of the VEGFA/NF-κB pathway is essential for angiogenesis (Lin et al., 2023; Mantsounga et al., 2022). In this study, miR-381-3p mimics significantly reduced Vegfa mRNA expression in C166 cells (P < 0.01), whereas miR-381-3p knockdown increased Vegfa mRNA expression (P < 0.001) (Figs. 3A and 3B). Notably, miR-381-3p mimics inhibited the expression levels of the VEGFA and p-p65 proteins (Fig. 3C). These findings indicated that miR-381-3p may inhibit angiogenesis in C166 cells by regulating the activation of the VEGFA/NF-κB pathway.

Figure 3 miR-381-3p inhibited the activation of VEGFA/NF-κB pathway.

(A) miR-381-3p expression was measured using RT-qPCR. (B) The Vegfa mRNA expression levels were detected using RT-qPCR. (C) The protein levels of the VEGFA, p65, and p-p65 were analyzed using western blotting. The full-length gels and blots of the proteins are included in Fig. S1. Data are presented as mean ± SD (n = 3). **P < 0.01 and ***P < 0.001.

miR-381-3p suppressed the migration and angiogenesis of C166 cells through targeting VEGFA

As presented in Fig. 4A, a dual-luciferase reporter system was used to identify the binding of miR-381-3p and VEGFA 3′ UTR. The results demonstrated that miR-381-3p mimics significantly decreased the relative fluorescence activity in the VEGFA-WT group but had no effect on the fluorescence activity in the VEGFA-MUT group (Fig. 4A). Subsequently, three siRNA sequences of VEGFA were constructed. RT-qPCR verification revealed that VEGFA siRNA#2 most significantly decreased Vegfa mRNA expression (Fig. 4B). As a result, VEGFA siRNA#2 was used to silence Vegfa mRNA expression. VEGFA silence reversed the effect of the miR-381-3p inhibitor on C166 cell migration and angiogenesis. These results revealed that miR-381-3p inhibited the migration and angiogenesis of C166 cells by targeting VEGFA.

Figure 4 Effect of miR-381-3p on the migration and angiogenesis of C166 cells via targeting VEGFA.

(A) The dual-luciferase reporter system was used to verify the binding of miR-381-3p to VEGFA 3' UTR. (B) The interference efficiency of the VEGFA siRNA was detected by RT-qPCR. (C) Migration and angiogenesis of C166 cells were measured using Transwell (magnification: 20×), wound healing (magnification: 10×), and tube formation (magnification: 10×) assays. (D) Migration number of C166 cells. (E) Migration distance of C166 cells. (F) Branch points in angiogenesis of C166 cells. (G) Capillary length in the angiogenesis of C166 cells. Data are presented as mean ± SD (n = 3). *P < 0.05 and ***P < 0.001.

miR-381-3p knockdown partially rescued abortion by regulating vascular remodeling, inflammation, and RSA-related factors

To explore the role of miR-381-3p in the placenta and fetus in the abortion mouse model, AAV-mediated shmiR-381-3p was constructed and injected intraperitoneally into pregnant mice to achieve miR-381-3p knockdown. In the control group, the placenta and fetus were intact; however, the abortion mice had fetal deformities and lower weights of the placenta and fetus (Fig. 5A). miR-381-3p knockdown improved the fetal status of the abortion model, including observable limbs and increased weight of placenta and fetus, compared with the abortion group (Fig. 5A). Moreover, CD31 positive expression in the abortion group was significantly lower than that in the control group (P < 0.01), while CD31 expression in the abortion+shmiR-381-3p group was significantly higher than that in the abortion group (P < 0.05) (Fig. 5B), suggesting that miR-381-3p knockdown may promote angiogenesis in the placenta to improve ischemic/anoxic state of the placenta in the abortion mice. HE staining results revealed that the decidual cells were abundant, with scattered inflammatory cells; the vascular structure was normal, and trophoblast cells were visible without vascular necrosis. In the abortion group, the decidual cells decreased, and the blood vessel wall thickened, accompanied by vascular necrosis. In the abortion+shmiR-381-3p group, the decidual cells were abundant, and vascular thickening or necrosis was not observed (Fig. 5C).

Figure 5 miR-381-3p knockdown partially alleviates spontaneous abortion in mice.

(A) Representative images of the placenta and fetus in each group were presented, and the weights of the placenta and fetus were measured (n = 5). (B) The CD31 expression was analyzed using IHC staining (n = 3). Magnification: 10×. (C) Morphological changes in the placentas were observed by HE staining (n = 3). Magnification: 40×. ns P > 0.05, *P < 0.05, **P < 0.01, and ***P < 0.001.

Compared with the control group, the expression level of Vegfa mRNA decreased significantly in the abortion group (P < 0.001), whereas miR-381-3p knockdown reversed this result (Fig. 6A). The expression levels of VEGFA and p-p65 proteins decreased significantly in the abortion group compared with the control group (P < 0.01), while miR-381-3p knockdown increased the expression levels of VEGFA and p-p65 proteins than the abortion group. The levels of IL-10, ANX-V, and MTHFR in the abortion group were significantly lower than those in the control group (P < 0.001), whereas miR-381-3p knockdown significantly increased the levels of the above three factors, compared with the abortion group (P < 0.01) (Figs. 6C–6E). TNF-α and HCY levels revealed opposite results (Figs. 6F and 6G). Elevated HCY level induces endothelial cell damage to increase uterine artery resistance in the non-pregnant state, which leads to an increased risk of RSA (Yang et al., 2023). ANX-V can maintain a normal pregnancy by preventing blood clotting and may play a beneficial role in RSA (Murad et al., 2023). Furthermore, the inactivation or absence of MTHFR leads to HCY accumulation, increasing the risk of RSA (Lei et al., 2024; Zhang, Fu & Wei, 2021). These findings suggest that miR-381-3p knockdown can rescue abortion by promoting angiogenesis, inhibiting inflammation, and regulating RSA-related factors.

Figure 6 miR-381-3p knockdown improved inflammation in the abortion mouse model.

(A) The expressions of the Vegfa mRNA were detected by RT-qPCR. (B) The protein expressions of the VEGFA, p65, and p-p65 were analyzed using western blotting. The full-length gels and blots of the proteins are included in Fig. S2. (C–G) ELISA was used to detect the levels of inflammatory and abortion-related factors. Data are presented as mean ± SD (n = 3). *P < 0.05, **P < 0.01, and ***P < 0.001.

Discussion

RSA is the most common pregnancy-related complication and affects 1–5% of reproductive-age women (Mou et al., 2022; Sonehara et al., 2024). There is increasing evidence that angiogenesis is critical for establishing a successful pregnancy (Liu et al., 2022; Murata, Tanaka & Okada, 2022; Sandovici et al., 2022). Dysfunctional endothelial cells cause the placenta to be in a state of hypoxia/ischemia (Tan et al., 2024), increasing the abortion risk. In our study, miR-381-3p was highly expressed in the placental tissue of the abortion mouse model and bound to VEGFA. miR-381-3p overexpression inhibited C166 cell migration and angiogenesis by targeting the VEGFA/NF-κB pathway, whereas miR-381-3p knockdown had the opposite effect. VEGFA siRNA reversed the positive effect of miR-381-3p knockdown on the angiogenesis of C166 cells. An in vivo experiment further demonstrated that miR-381-3p knockdown rescued abortion via regulating vascular remodeling and inflammation.

RSA has always been a problem worldwide, with nearly 50% of patients unable to identify the cause. The known causes mainly include endocrine, genetic disorders, autoimmune diseases, and anatomical abnormalities (Davalieva, Kocarev & Plaseska-Karanfilska, 2025; Han et al., 2025). Many studies have confirmed that miRNAs are involved in the RSA process (Xiong et al., 2024; Zhao et al., 2022). miRNAs regulate multiple biological functions, such as cell differentiation, development, proliferation, and apoptosis, by degrading target genes or inhibiting their protein translation (Diener, Keller & Meese, 2024; Saliminejad et al., 2019). Recently, miRNA has gradually become the “star molecule” in the diagnosis and treatment of pregnancy syndrome (Jaszczuk et al., 2022; Liang et al., 2023; Mavreli, Papantoniou & Kolialexi, 2018; Srivastava et al., 2022). During pregnancy, placental trophoblasts produce numerous miRNAs (Liang et al., 2023). These miRNAs have unique communication patterns between the mother, placenta, and fetus and regulate trophoblastic functions in multiple aspects, including proliferation, invasion, and metabolism (Hemmatzadeh et al., 2020; Liang et al., 2023; Lip et al., 2020). In this study, miR-381-3p knockdown was beneficial in promoting angiogenesis in vivo and in vitro, maintaining placental and fetal integrity in the abortion mice, and inhibiting inflammatory response and production of abortion-related indicators, indicating that miR-381-3p may contribute to the pathogenesis of RSA.

Recent evidence has suggested that VEGFA dysregulation is a crucial factor in RSA (Krog et al., 2024; Liao et al., 2024). VEGFA expression was negatively correlated with fetal growth restriction (Pei et al., 2022). VEGFA, a multifunctional factor, regulates endothelial cell proliferation and migration, angiogenesis, and vascular permeability (Apte, Chen & Ferrara, 2019; Shaw et al., 2024). VEGFA levels increase during the middle luteal phase (Sugino et al., 2002), suggesting that VEGFA is a potential predictor of successful embryo implantation (Alves, Dias & Silvestre, 2023). Contrarily, VEGFA deficiency leads to adverse events during pregnancy (Hannan et al., 2011). Blocking the VEGFA signaling pathway has been demonstrated to lead to decreased detachment of vascular endothelial cells, arrest of embryonic development, and preterm birth (Shi et al., 2023; Smithmyer et al., 2021; Wada et al., 2013).

It has been reported that the VEGFA expression decreases in the endometrial tissue of patients with idiopathic recurrent abortion during the peri-implantation period, which may be related to the downregulation of angiogenic cytokines, including IL-2, IL-6, and IL-8 (Banerjee et al., 2013). Similarly, VEGFA expression levels are low in the endometrium during the middle luteal phase in women with RSA (Sadekova et al., 2015). VEGFA also contributes to embryo implantation by stimulating endometrial angiogenesis (Guo et al., 2021). In our study, miR-381-3p suppressed C166 cell migration and angiogenesis by targeting VEGFA. Notably, miR-381-3p knockdown can promote placental vascular remodeling to improve abortion via increased VEGFA expression and reduced inflammation. Interestingly, we found that miR-381-3p knockdown activated the NF-κB pathway, regulated the expression of RSA-related indicators, and improved placental damage in the abortion mice. It is generally believed that NF-κB pathway activation promotes inflammation (Cornice et al., 2024; Yu et al., 2020), but some studies have suggested that NF-κB pathway is vital for neovascularization (Lin et al., 2023; Mantsounga et al., 2022). Consequently, NF-κB pathway activation regulated by miR-381-3p may mediate angiogenesis rather than inflammation; however, the specific reasons for this need to be further explored.

In conclusion, the study confirmed that miR-381-3p regulated the angiogenesis of C166 cells by targeting the VEGFA/NF-κB pathway. miR-381-3p knockdown had a protective effect on the abortion mice, such as promoting placental tissue angiogenesis, maintaining fetal mouse integrity, inhibiting inflammation, and regulating abortion-related indices (Fig. 7), suggesting that miR-381-3p can be used as a new therapeutic target for RSA. However, this study only verified the effect of miR-381-3p on abortion in vitro and in vivo, and the validity of the findings was limited by a small sample size. The effect of miR-381-3p on the prevention, diagnosis, and treatment of RSA in the clinic requires further research. Furthermore, there is a lack of studies on the interaction between miR-381-3p and VEGFA in vivo. In the future, we will expand the replication of results in animal models, human placental samples, or primary cell cultures to validate the findings further and focus on the deeper mechanism of miR-381-3p targeting VEGFA to control RSA.

Figure 7 Molecular pattern of miR-381-3p contribution in mouse spontaneous abortion via targeting VEGFA.

VEC, vascular endothelial cell; TNF-α, tumor necrosis factor-α; IL-10, interleukin-10; HCY, homocysteine; ANX-V, annexin V; MTHFR, 5,10-methylenetetrahydrofolate reductase. The pattern was created using FigDraw.

Supplemental Information

Supplemental Information 1 Tube formation assay of C166 cells in Figure 2A and Figure 4C.

Supplemental Information 2 Wound healing analysis of C166 cells migration in Figure 2A and Figure 4C.

Supplemental Information 3 Transwell analysis of C166 cells migration in Figure 2A and Figure 4C.

Supplemental Information 4 RT-qPCR analysis of miR-381-3p expression levels in C166 cells after miR-381-3p mimics or inhibitor.

Supplemental Information 5 RT-qPCR analysis of VEGFA mRNA expression in C166 cells after miR-381-3p mimics or inhibitor.

Supplemental Information 6 The full-length uncropped gels/blots of VEGFA, p65, and p-p65 protein expressions in C166 cells.

Supplemental Information 7 VEGFA mRNA expression in C166 cells after VEGFA siRNA.

Supplemental Information 8 Original picture of placenta and fetus in Figure 5 A.

Supplemental Information 9 VEGFA mRNA expression in the placentas.

Supplemental Information 10 The full-length uncropped gels/blots of VEGFA, p65, and p-p65 protein expressions in the placentas.

Supplemental Information 11 MIQE checklist.

Supplemental Information 12 Author Checklist.

Supplemental Information 13 Supplementary materials.

We would like to acknowledge Nanjing Ramda Pharmaceutical Co., Ltd. for conducting the animal testing.

Additional Information and Declarations

Competing Interests

The authors declare that they have no competing interests.

Author Contributions

Chao Ding conceived and designed the experiments, authored or reviewed drafts of the article, and approved the final draft.

Fukang Liu performed the experiments, prepared figures and/or tables, and approved the final draft.

Huayue Shi analyzed the data, prepared figures and/or tables, and approved the final draft.

Jing Zuo analyzed the data, authored or reviewed drafts of the article, and approved the final draft.

Lei Bi conceived and designed the experiments, prepared figures and/or tables, and approved the final draft.

Longgang Shao analyzed the data, authored or reviewed drafts of the article, and approved the final draft.

Yanqiu Pan conceived and designed the experiments, performed the experiments, prepared figures and/or tables, authored or reviewed drafts of the article, and approved the final draft.

Animal Ethics

The following information was supplied relating to ethical approvals (i.e., approving body and any reference numbers):

The animal experiment was approved by the Institutional Animal Care and Use Committee of Nanjing Ramda Pharmaceutical Co., Ltd. (No. IACUC-20230208) and performed per the Animal Care Committee.

Data Availability

The following information was supplied regarding data availability:

The raw measurements are available in the Supplemental Files.

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
