# Peer review of "miR-381-3p contribution in mouse spontaneous abortion via targeting VEGFA"

_PeerJ, doi:10.7717/peerj.19568_

## Round 0.1 · original submission · Major Revisions

· Academic Editor

Major Revisions

Please address concerns of both reviewers and revise manuscript accordingly.

Reviewer 1 ·

Basic reporting

Clarity and Language:
• The manuscript is well-written and clear. The scientific terms and technical language are appropriately used, making the article accessible to a wide academic audience while maintaining technical precision.
Literature References and Background:
• The authors provide sufficient background on the role of angiogenesis, miRNAs, and Recurrent spontaneous abortion (RSA). They have appropriately referenced previous studies that are relevant to the research question. I find the literature review adequate, though there is a notable reliance on older sources when discussing the role of VEGFA and NF-κB in RSA. Recent studies with recent references (2023-2024 publication references) on miRNA regulation of angiogenesis in pregnancy could enhance the manuscript’s relevance to current research trends.
Structure:
• The manuscript is well-structured. Figures and tables are clear and well-organized. The data are presented well.
Self-contained:
• The manuscript is self-contained, with the results supporting the hypotheses. The hypothesis and research question are well-defined, and the results directly address the proposed objectives.
Suggestions for Improvement:
For Western Figures 2 and 5 (I would appreciate the ladder image along with its molecular weight (I see the molecular weight; however, the ladder image in it will show transparency).
In addition, the sample size should be included in the figure legend.

Experimental design

Originality and Relevance:
• The study presents original research that fits well within the aims and scope of the journal. The research question is well-defined.
Filling the Knowledge Gap:
• The authors effectively fill an identified knowledge gap by investigating how miR-381-3p regulates angiogenesis in the context of RSA. While previous studies have explored the role of VEGFA in RSA, this work introduces miR-381-3p as a key regulator, which has significant implications for understanding placental repair and fetal development during pregnancy.
Technical and Ethical Standards:
• The experimental design adheres to high technical standards, with appropriate controls, statistical analyses, and validation experiments. The use of C166 cells for angiogenesis assays and in vivo testing in a mouse model of RSA is well-suited to the research question. However, while the authors detail the experimental protocols clearly, there is limited information on sample sizes or statistical power calculations, which could strengthen the validity of the findings.
Methods and Replicability:
• The methods are described in sufficient detail to allow replication. However, providing more information on the in vivo model (e.g., which cycle were the mice during experiments? As different phases (Estus cycling) make a huge difference. In addition, the exact gestational age when miR-381-3p knockdown is performed) would make the experiments easier to reproduce. Additionally, while the authors mention using a dual-luciferase reporter system, further clarification on how this assay was set up (e.g., specific regions of VEGFA’s 3' UTR tested) and complete details would benefit the reader.

Validity of the findings

Impact and Novelty:
• The findings are new and have the potential to make a significant impact on the field of RSA research. miR-381-3p is shown to regulate VEGFA expression and angiogenesis in the context of RSA, offering a new molecular target for therapeutic intervention.
Replication:
• The study provides initial evidence of the effects of miR-381-3p modulation in C166 cells and mice. It would be beneficial to replicate these results meaningfully, both in additional in vivo models and in human tissues. The authors could consider expanding the replication of results in human placental samples or primary cell cultures to validate the findings further.
Statistical Soundness:
• The authors provide appropriate statistical analyses, and the data are robust and statistically sound in the figures. However, there is a lack of detail about the sample size calculations or how the power of the experiments was determined. Including this information would strengthen the validity of the results.
Conclusions:
• The conclusions are well-stated and linked to the original research question. The authors conclude that miR-381-3p is involved in regulating angiogenesis via VEGFA, and this could be a potential therapeutic target for RSA. The findings are well-supported by the data, and the conclusions are appropriately conservative and not overstated.

Additional comments

This manuscript presents promising findings regarding miR-381-3p as a potential therapeutic target for RSA. While the experimental design is rigorous, and the data are promising, some areas could be strengthened, particularly regarding sample size, mouse phase during the experiment (estrus cycling), and statistical power.

Reviewer 2 ·

Basic reporting

In the paper: miR-381-3p contribution in mouse spontaneous abortion via targeting VEGFA, the authors were interested in evaluation of the contribution of a specific miR-381-3p in the progression of recurrent spontaneous abortion (RSA). To do so, they report on the construction of a mouse model of RSA and use molecular and histological techniques to gain information on their research questions. The data presented are of interest to the research community and present new evidence in the field, however there are a few major concerns to be addressed prior to its consideration for publication.

Experimental design

- It is not clear why the authors only choose one single miR to investigate as potential regulator of RSA, when they have done a transcriptome sequencing of aborted placenta and have thus generated the whole panel of expressed miRs in this mice tissue?
- Why was the mouse model β2-GPI used, knowing that this model is actually used to study the autoimmune aspects contributing to RSA and in relation to antiphospholipid antibodies?

Validity of the findings

The authors must provide the experimental RNASeq description and a validation of the seleceted miRNAs by qPCR.
If the authors used transcriptome sequencing, they must provide evidence of the RNASeq data, at the MM authors must explain how the RNASeq was performed, and the data must be deposited at the ENA, GEO, or other databases. It is important to show if other miRs, either as part of the miR-381 cluster or other unrelated miRs are among the differentially expressed miRNA.

Additional comments

Other minor comments:

Abstract

Line 40: the word improved damage is not suitable.


Introduction

The introduction is weak on evidence and needs to be improved, there are a lot of sentences without reference e.g. line 63-69.

Line 94: please give the reference, which database and how strong the prediction of MTI was (miR-381:VEGFA).

Material and Methods


Line 113: “Three biologically in each group were performed”, revise the sentence, not clear what was done?

Results

Overall the results are not well structured, and often also discussed at the same time, which should not be the case
Line 242: please give the references for those databases.
Line 266: here also references are missing.
Line 267: a lot of typos, also throughout the text, please correct.


Discussion

Line 327-330: please give the reference and this follows the rest of the discussion.

---

## Round 0.2 · Major Revisions

· Academic Editor

Major Revisions

Please address the remaining issues pointed out by the reviewer and revise the manuscript accordingly.

**Language Note:** The review process has identified that the English language must be improved. PeerJ can provide language editing services - please contact us at [email protected] for pricing (be sure to provide your manuscript number and title). Alternatively, you should make your own arrangements to improve the language quality and provide details in your response letter. – PeerJ Staff

Reviewer 1 ·

Basic reporting

The authors have addressed the previous issue with the article, and the article looks better now

Experimental design

The experimental design adheres to high technical standards, with appropriate controls, statistical analyses, and validation experiments.
Previous issues have been addressed.

Validity of the findings

Previous issues have been addressed.

Additional comments

This manuscript presents promising findings regarding miR-381-3p as a potential therapeutic target for RSA

Reviewer 2 ·

Basic reporting

The manuscript must be checked for professional English, as there are plenty of typos and errors e.g. line 307 "gourps".

Experimental design

Although the authors have addressed most of the comments from the first round of review, it remains unclear whether an RNASeq experiment was conducted as part of this study or if the authors are referring to RNASeq results from the study by Li et al. (2024). This point needs to be clarified for the reader. If RNASeq was indeed performed, the annotated RNA reads must be deposited in a repository, e.g., ENA

Validity of the findings

no comment

---

## Round 0.3 · accepted · Accept

· Academic Editor

Accept

All remaining issues were addressed, and the revised manuscript is acceptable now.

Reviewer 2 ·

Basic reporting

The authors have thoroughly addressed all the concerns raised in my review. I therefore recommend the paper for publication.

Experimental design

The authors have thoroughly addressed all the concerns raised in my review. I therefore recommend the paper for publication.

Validity of the findings

The authors have thoroughly addressed all the concerns raised in my review. I therefore recommend the paper for publication.